

# Optimizing lettuce yields and quality by incorporating movable downward lighting with a supplemental adjustable sideward lighting system in a plant factory

Mulowayi Mutombo Arcel[1,2], Ahmed Fathy Yousef[3],
Zhen Hui Shen[1,2,4], Witness Joseph Nyimbo[5] and Shu He Zheng[1,2]

[1] College of Mechanical and Electrical Engineering, Fujian Agriculture and Forestry University, Fuzhou, Fujian, China
[2] Fujian University Engineering Research Center for Modern Agricultural Equipement, Fuzhou, Fujian, China
[3] Department of Horticulture, College of Agriculture, University of Al-Azhar (Branch Assiut), Assiut, Egypt
[4] Engineering College, Fujian Jiangxia University, Fuzhou, Fujian, China
[5] Fujian Provincial Key Laboratory of Agro-Ecological Processing and Safety Monitoring, College of Life Sciences, Fujian, Agriculture and Forestry University, Fuzhou, Fujian, China

Corresponding author
Shu He Zheng, zsh@fafu.edu.cn

## ABSTRACT

**Background:** Lettuce is a vegetable that is increasingly consumed globally, given its nutritional quality. Plant factories with artificial lighting can produce high-yield and high-quality plants. High plant density in these systems speeds up leaf senescence. Wasted energy and lower yield raised labor expenses are some of the bottlenecks associated with this farming system. In order to increase lettuce yields and quality in the plant factory, it is essential to develop cultivating techniques using artificial lighting.

**Methods:** Romaine lettuce was grown under a developed "movable downward lighting combined with supplemental adjustable sideward lighting system" (C-S) and under a system without supplemental sideward lighting (N-S) in a plant factory. The effects of C-S on lettuce's photosynthetic characteristics, plant yield, and energy consumption relative to plants grown under a system without N-S were studied.

**Results:** Romaine lettuce growth and light energy consumption in the plant factory were both influenced favorably by supplementary adjustable sideward lighting. The number of leaves, stem diameter, fresh and dry weights, chlorophyll $a$ and $b$ concentration, and biochemical content (soluble sugar and protein) all increased sharply. The energy consumption was substantially higher in the N-S treatment than the C-S.

## INTRODUCTION

The primary global problems of the twenty-first century are food, resources, the environment, and the quality of life. Human activities, which also lead to soil compaction,

soil salinization, heavy metal toxicity, extreme pollution, and desertification cause the natural environment to deteriorate (*Arcel et al., 2021*). Moreover, climate change's increased complexity associated with agricultural production has made predictable and consistent yields more difficult (*Malhi, Kaur & Kaushik, 2021*). With the help of closed-environment methods like plant factories, vegetables can be effectively cultivated under challenging conditions. Plant factories can grow high-yield, high-quality plants with less water, and less fertilizer inputs than that of conventional agriculture. Plant factories are developing industries with the potential to address some of these conundrums (*Kozai, 2013a*; *Hu, Chen & Huang, 2014*). To implement highly precise control of environmental parameters in such growth environment, the plant factory needs to be integrated with contemporary industry, biotechnology, nutrient solution cultivation, and information technology (*Kozai, 2019*). These are necessary for plants to grow entirely or partially without being affected by its environment (*Mitchell, 2012*). Regarding the latter, it alludes to the so-called "plant factories" that employ solar lighting to help producers produce large yields.

Light is one of the most critical environmental factors affecting plant growth and development (*Ahmed, Yu-Xin & Qi-Chang, 2020*). It is an important energy source for plant photosynthes is, and can affect morphogenesis and material metabolism of plants (*Elmardy et al., 2021*; *Liang et al., 2021*). However, it is challenging to control natural light due to many factors, including geographical location, meteorological conditions, *etc.* (*Knoop et al., 2019*; *Cianconi, Betrò & Janiri, 2020*). Hence, it is not an ideal light source for studying plant light regulation and plant growth regulation (*Kami et al., 2010*). Nevertheless, source such as artificial lights have proven to be suitable for such study, although it can not substitute the sun completely. The fluorescent light is usually used as a light source for indoor horticultural production, including micropropagation. Studies have shown that light emitting diodes (LEDs) has the advantages of low heat generation, long service life, good spectral performance, and easy control (*Amoozgar, Mohammadi & Sabzalian, 2017*; *Ferreira et al., 2017*). In plant factory research, the adoption artificial light sources such as LEDs have proven to have great potential in the regulation on plants growth development (*Dutta Gupta & Agarwal, 2017*; *Efremova et al., 2020*). In addition, LED can exhibit the advantage of inducing a significant change in plant growth, especially in closed plant factories (*Saito, 2010*), while significantly increasing the area under cultivation of plants and reducing the shortage of farm land resources (*Kozai, 2013b*). Furthermore, artificial light sources are used to provide a bright background for plant growth and development (*Dutta Gupta & Agarwal, 2017*; *Efremova et al., 2020*).

Although artificial light sources serve as a good source of plant regulation in plant factory, obtaining a quality crop production remains a significant challenge partly due the continuously changing plant's light absorption area (*Proietti et al., 2021*; *Yang, Song & Jeong, 2022*). Moreover, in conventional plant lighting systems, artificial light sources and irradiation areas are fixed (*Dutta Gupta & Agarwal, 2017*; *Efremova et al., 2020*), which may also result in noticeable differences in light throughout the cultivation area based on the light properties of artificial light sources. Therefore, research into a light control device and its control system based on the illuminance distribution properties of artificial light

sources, which can dynamically adjust the lighting conditions of the plant's light absorption area in real-time is essential (*Carvalho & Folta, 2014*).

Although uniform illumination is desirable, optimum energy consumption is a significant factor in designing LED light sources (*Wu & Gao, 2018*). The development of intelligent machine vision systems has demonstrated effectiveness for accurately controlling plant moisture content and lighting intensity through out the growth and development of plants (*Hendrawan, Al Riza & Murase, 2014*). Also, these systems can optimize plant growth and reduce water consumption and energy costs. However, light regulation remains a significant issue in large-scale plantations (high plant density) due to occlusion caused by adjacent plant leaves. The occlusion makes it difficult for the middle and lower leaves to receive adequate light compared to the upper leaves (*Terashima et al., 2005*). As a result of the leaf area, light absorption through the leaf surface is regulated, along with leaf morphology and orientation (*Yang, Song & Jeong, 2022*). For instance, tomatoes treated with red LEDs caused the tomato leaves to curl upward or downward. However, anomalies in the leaf morphology of the tomatoes were reduced, and the plant biomass peaked when treated with red and blue LED (*Ouzounis et al., 2016*). Moreover, the abaxial side of the epidermal cells in geranium elongated in response to directional blue light irradiation in a red-light environment, which inhibited the growth of new leaves (*Fukuda et al., 2008*). In related studies, it was documented that irradiation of a leaf's adaxial and abaxial sides increased photosynthesis (*Terashima, 1986*; *Soares et al., 2008*), and different light colors exhibited different effects on leaf senescence (*Causin, Jauregui & Barneix, 2006*). Likewise, lettuce growth, quality, and light optimization in plant factories have been the subject of numerous studies (*Joshi et al., 2017*). However, no studies have been conducted to investigate an ingenious way to boost lettuce yields and quality in a plant factory by incorporating movable downward lighting with adjustable sideward lighting. To fill this knowledge gap, a smart system was developed to help farmers boost lettuce yields and quality in a plant factory by combining movable downward lighting with a supplemental adjustable sideward lighting system, while delaying senescence, providing high-quality light but maintaining low output intensity. The system's advantages were assessed by looking at the lighting's electricity consumption, plant growth, plant physiology, and the impact of the C-S on leaf senescence.

## MATERIALS AND METHODS

### Design of plant light source regulation experiment
#### *Overall design of mechanical structures*
The primary design and function of the plant light control device are as follows: (1) Adjust the distance of the plant light source and the plant height in a better illumination region. (2) To realize the detection of plant cultivation location. (3) Complete plant growth height detection. (4) Turn on the supplemental sideward plant light source and adjust the angle for a better illumination region. (5) Adjust the lighting area of the light source so that the plant cultivation position is in the optimal light region. (6) Light quality and intensity can be adjusted to meet the plant growth needs at all stages. (7) Human-computer interaction.

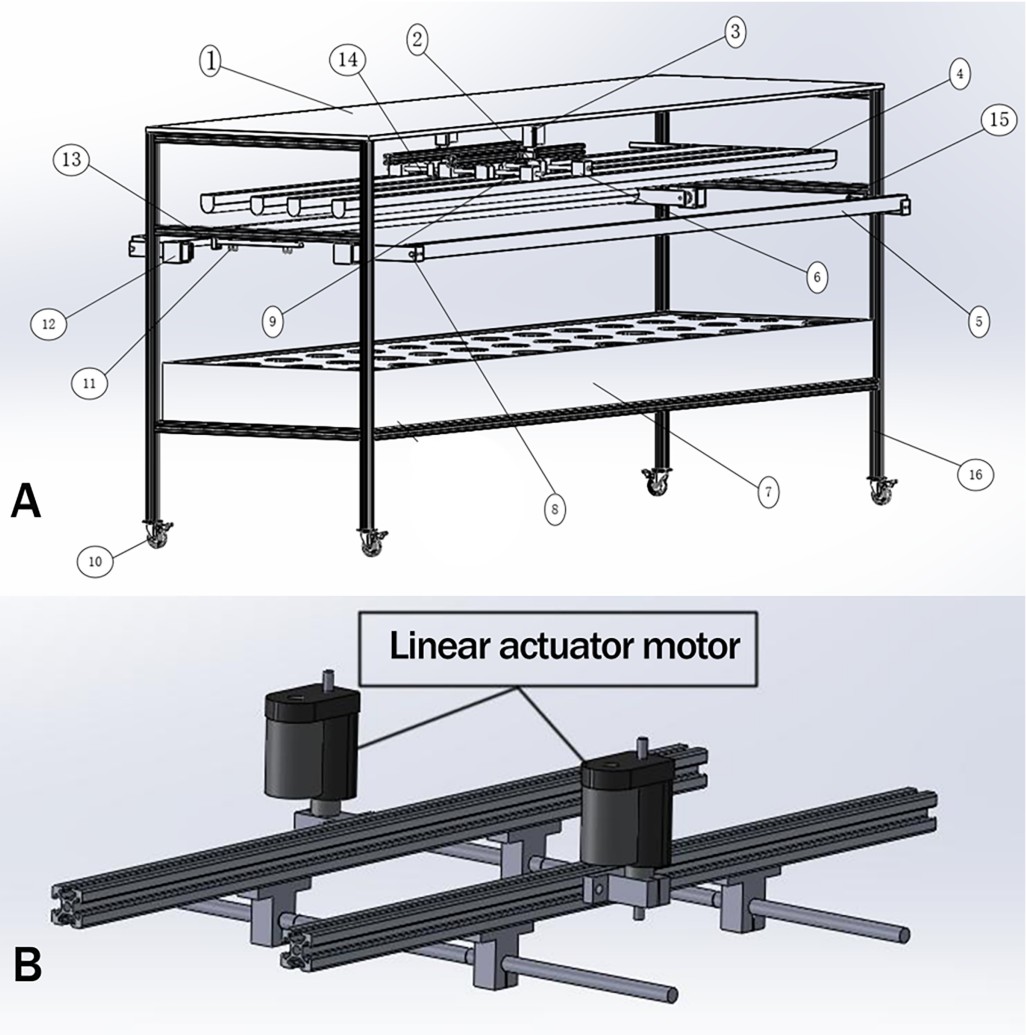

**Figure 1 The overall structure of the light control device.** The primary design function of the plant light control device is as follows: (1) adjust the distance of the plant light source, and the plant height in a better illumination region. (2) To realize the detection of plant cultivation location. (3) Complete plant growth height detection. (4) Turn on the supplemental sideward plant light source and adjust the angle for a better illumination region. (5) Adjusting the lighting area of the light source so that the plant cultivation position is in the optimal light region. (6) The light quality, light intensity can be adjusted to meet the plant growth needs at all stages. (7) Human-computer interaction. The above design requirements, (1), (2), and (5) need to be realized by mechanical structure and control system; (3), (4), (6), and (7) can be done by the control system. Taking the light control device design of the ONE-layer plant as an example, its overall structure is shown in (A), and the downward mechanism is shown in (B).

The above design requirements, (1), (2), and (5) need to be realized by mechanical structure and control system, (3), (4), (6), and (7) can be done by the control system. Taking the light control device design of the ONE-layer plant as an example, its overall structure is shown in Fig. 1.
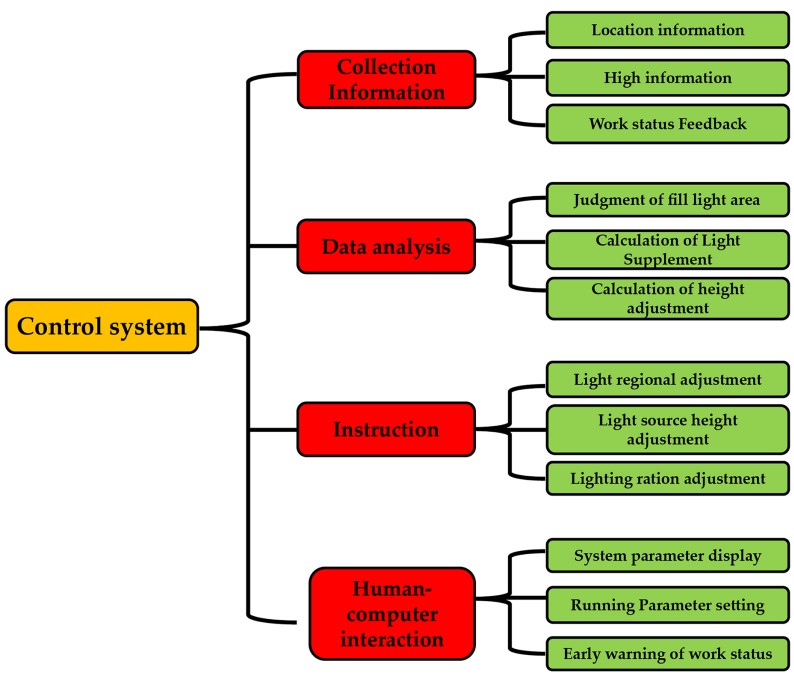

**Figure 2  System function breakdown diagram.** 

## System function analysis

In this study, the system's main functions were divided into four functional modules: a collection of information, decision analysis, instruction execution, and human-computer interaction, as shown in Fig. 2. According to the plant light control device's design requirements, the control system's functions were designed as follows: The information collection function module includes information about plants' growing positions, the height of plant growth, and the working state. The decision analysis function module assesses the various light areas, the calculation of additional light, and the repositioning of the light source. The human-machine interaction functions module, which displays the system's operating parameters and provides an early warning of the system's operational status is defined by the system's operating parameters. This function module executes instructions, including adjusting the installation's lighting area, lighting ratio, and light source height.

## Sideward LEDs control method

The system is controlled by a human-computer interface, which uses infrared sensors to gather distance data from the plant factory and a lighting sensor to measure ambient lighting. Next, it determines the distance between the plate, the LEDs, and the best pulse-width modulation (PWM) for the environmental photosynthetic photon flux density (PPFD) using the target PPFD. In this instance, the PWM signal managed the PPFD. As a result, after passing through the power adjustment module, which converts the PWM signal to DC power, the system generates the optimal level of the PWM signal while taking into account the distance between the center LED and the surrounding PPFD, as shown in Fig. 3.

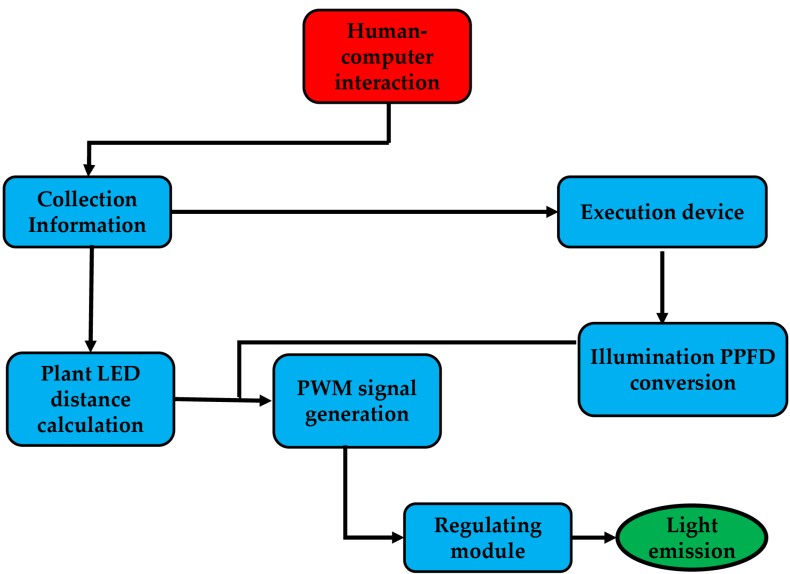

**Figure 3 The system runs from human-computer interaction.**

### System overall scheme design

The detection module consists of two parts: the installation position detection module and the installation height detection module. The execution module consists of the sensor position adjustment module, the light source height module, and the light module, as shown in Fig. 4.

The power module adopts a multi-voltage output switching power supply to provide power protection for the remaining modules of the control system. The control module adopts the Arduino Mega 2560 (Shenzhen Ke Zhi You Technology co., Ltd, Shenzhen, China) as the core controller, and the control system achieves the expected function. The plant position detection module uses a micro switch to switch, and the matrix switch is responsible for collecting plant cultivation position information and transmitting it to the control module for processing. The plant height detection module consists of ultrasonic sensor modules, and the adjustment module consists of a driver, a stepping motor, and a screw mechanism. The light source height adjustment module adjusts the distance between the LED panel light source and the plant, and the interactive module is composed of LCD1602 display and setting buttons. The early warning module includes an indicator light and a buzzer to indicate the working status of the system and alarm when the system works abnormally.

### Light height adjustment module for plant

An air guide motor's driver and a screw rod structure are part of the light height adjustment module connected to the upper computer controller. Changzhou Longxiang Company's electric linear actuator motor was adopted for adjusting the height of the downward LED lighting sources. A BXTL 150 electric linear actuator motor with a 100 mm pushrod stroke, 100 kg thrust, and a rated speed of 12 mm s$^{-1}$ was employed to move
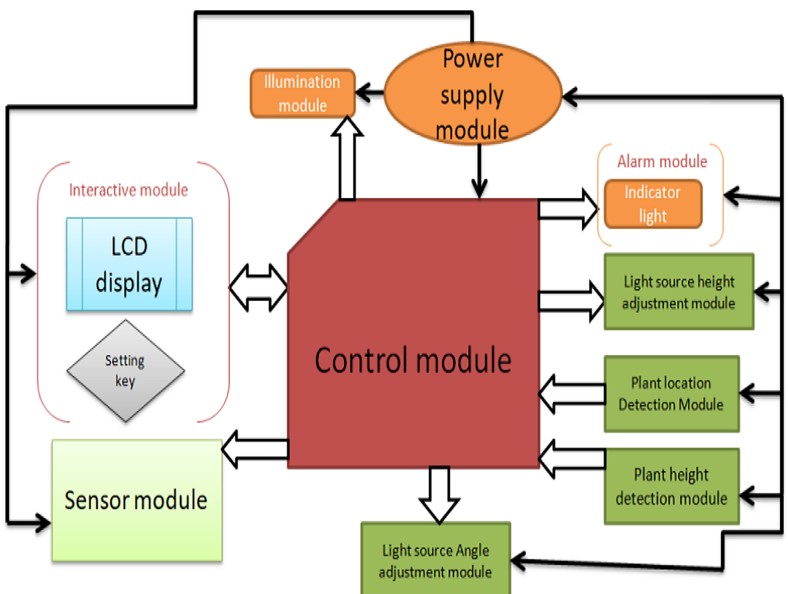

**Figure 4 The overall structure of the system.**

downward lighting. Real-time measurements of the linear actuator motor's telescopic length and corner data are obtained by the incremental Hall encoder inside the push rod. The circuit design changes the distance between the installation and the LED light source to maintain a better range distance between the light source and the plant. The TB6560 stepper motor control board was employed as the drive because it has an automatic half current and can handle the large load and required drive current of the electrochemical stepper drive in this module. It has a high-stability, affordable stepper motor driver with reliability and anti-interference. It is suitable for various industrial control environments by utilizing high-speed optical isolation, over-current protection, and over-temperature protection functions.

*Growth conditions*

This experiment was conducted in the Key Laboratory of Crop Biotechnology, Fujian Agriculture and Forestry University. The hydroponic seedling cultivation method cultivated roughly 200 romaine lettuce seeds in a plastic cultivation box (110 mm × 235 mm × 40 mm). The seeds were placed on an open sponge block after adding clean water to ensure that the sponge submerged completely. After the lettuce seeds had sprouted, they were grown in an environment-controlled growth chamber at ~17–20 °C (photoperiod/dark period) for 12 h with a PPFD of 50 $\mu mol \cdot m^{-2} \cdot s^{-1}$ from cool white fluorescent lamps, and the water in the seedling box was changed to fresh nutrient solution. After every 12 h, a new nutrient solution was circulated for 15 min at a pH of 6.1 ± 0.1 and an EC of 1.6 ± 0.1 mS/cm. Following 4 weeks of water cultivation, 36 seedlings that consisted of leaf and of equal growth size were selected for the experiment. Among them 18 were chosen and grown under the C-S (Fig. 5), and the remaining lettuce seedlings were grown under the N-S. The cultivation board was constructed from a polyethylene
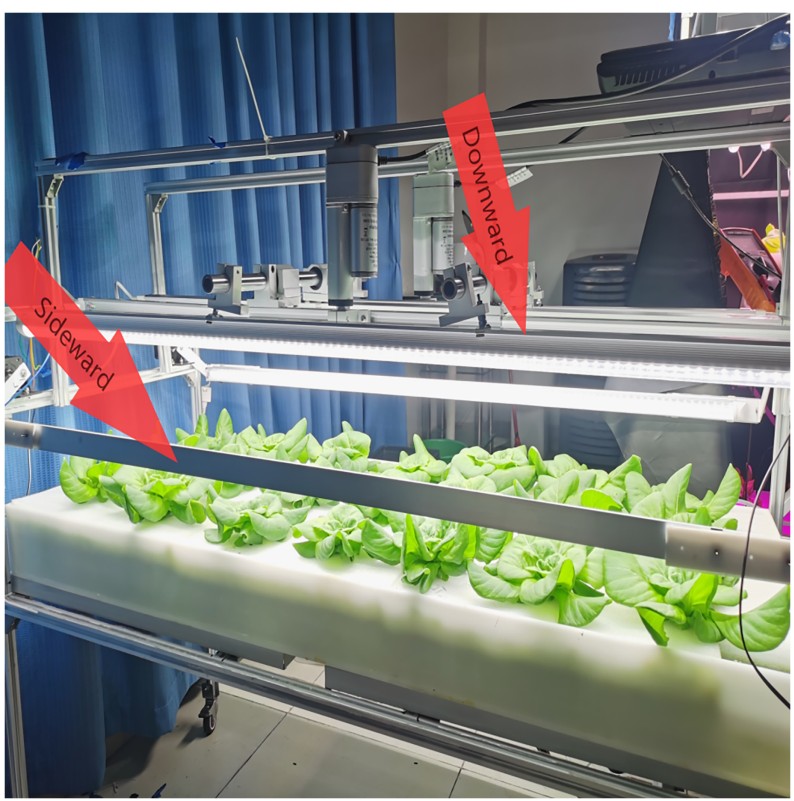

**Figure 5 The combination of the movable downward lighting and supplemental adjustable sideward LED lighting.**

foam board and measured 110 cm × 50 cm × 10 cm. The deep-flow hydroponics cultivation method was used, and the plant factory controlled several environmental factors during the cultivation period (*Causin, Jauregui & Barneix, 2006*). The air temperature at night was kept at 20.2 °C and 23 °C during the day. The relative humidity was kept at 60 ± 5%, and the carbon dioxide concentration was maintained at 400 ± 10 ppm. The factories were filled with pure $CO_2$ injected into the interior to ensure environmental controllability, and the carbon dioxide concentration was the same as the outside atmosphere. The transplanted seedlings were left in an improved fresh nutrient solution, and the pH value (pH = 6.5 ± 0.1 mS/cm) and EC value (EC = 1.7 ± 0.2 mS/cm) of the nutrient solution were regulated accordingly.

*Light treatments*

As seen in Fig. 5, the machine was constructed with a movable downward and flexible sideward lighting system. In total, six LED panels were used during the experiment for the two treatments C-S and N-S. A group without supplemental light was set as the control (N-S) with all six LED panels horizontally above the plant with the intensity of 280 $\mu mol \cdot m^{-2} \cdot s^{-1}$ at time 1, which is 20 days after transplanting (T1), and at time 2, 40 days after transplanting (T2), and there was no change. In the C-S, four LED panels were used for the moveable downward lighting system. These LED panels were placed horizontally above the plant with a PPDF of 200 $\mu mol \cdot m^{-2} \cdot s^{-1}$, two LED panels were

**Table 1 Lighting parameters and number of the LED source panel.** D + S (downward + supplemental sideward lighting) T1 (1 day transplanting to 20 after transplanting), T2 (20–40 days after transplanting).

| Parameter(s) | | (C-S) | (N-S) |
|---|---|---|---|
| | T1=(D+S) | 4 + 0 | 6 + 0 |
| Lighting source (LED panels) | | | |
| | T2=(D+S) | 4 + 2 | 6 + 0 |
| | T1=(D+S) | 200 + 0 | 280 + 0 |
| Light intensity (µmol m$^{-2}$ s$^{-1}$) | | | |
| | T2=(D+S) | 200 + 50 | 280 + 0 |
| Spectrum (nm) | | 360 to 760 | 360 to 760 |
| Time (hr) | | 8:00 to 24:00 | 8:00 to 24:00 |

placed on the side as supplementary adjustable sideward light. After monitoring multi-point averages in each treatment region following the plant growth, the additional adjustable sideward light angle was established. The supplemental adjustable sideward light intensity was controlled by regulating the lamps at 50 µmol·m$^{-2}$·s$^{-1}$ PPFD at T2 from 8:00 to 24:00 (Table 1). Spectroradiometers were used to identify the spectra of the light sources. A quantum radiation probe (FLA 623 PS; ALMEMO, Holzkirchen, Germany) was utilized to measure the additional light intensity at the top leaf level of the cuttings at night.

It should be noted that the sideward lights were turned on 20 days after transplanting, at which time all the outer leaves had taken on a shade-like appearance. The LEDs used were customized for the experiment by Sung Kwang LED Co., Ltd. in Incheon, Korea. The equipment produced a broad spectrum with a clear peak at 420 nm, ranging from 360 to 760 nm (blue). To control the electrical power, current, and PPFD of light sources, a switched-mode power supply was used.

### Plant growth and sample processing

Five plants from each replicate were sampled from T1 and T2 to explore their morphology parameters. Moreover, we measured the fresh weight (FW) and pigment content of the fresh lettuce plant. The shoot length, stem diameter and fresh and dry mass were collected as previously described in *Guiamba et al. (2022)*. According to the description of *Pandey & Singh (2011)*, the total leaf area (cm$^2$) (summation of leaf areas) was calculated. To determine the dry weight, fresh shoots and roots were placed in paper bags and placed in an oven set at 75 °C for at least 48 h.

### Photosynthetic pigments content

Chlorophyll content was extracted from the fresh lettuce leaves of both treatments. The fresh leaf tissue (0.2 g) was cut and ground thoroughly and later placed in 5 mL of 95% ethanol and filtered. The volume was then increased to 25 mL using 95% ethanol. A UV-5100B spectrophotometer was used to estimate the absorbance of the extracted solution at

665 nm (OD665), 649 nm (OD649), and 470 nm (OD470) (Unico, Shanghai, China). The chlorophyll content was calculated using the equations below (*Cataldo et al., 1975*; *Knight & Mitchell, 1983*):

$$\text{Chl } a \ (\text{mg g}^{-1}) = \frac{(13.95\text{OD}_{665} - 6.88\text{OD}_{649})\text{V}}{200\text{W}} \tag{1}$$

$$\text{Chl } b \ (\text{mg g}^{-1}) = \frac{(24.96\text{OD}_{649} - 7.32\text{OD}_{663})\text{V}}{200\text{W}} \tag{2}$$

$$\text{C } (\text{mg g}^{-1}) = \frac{(1000\text{OD}_{470} - 2.05\text{Chl}a - 114.08\text{Chl}b)\text{V}}{245 \times 200\text{W}} \tag{3}$$

where (Chl$a$) = chlorophyll $a$, (Chl$b$) = chlorophyll $b$, (C) = carotenoid, mg/g; (V) = volume (25 mL), and (W) = sample weight (g).

## Biochemical contents

### Nitrate content

Small pieces of fresh lettuce (0.6 g) were cut up, placed in a test tube with 10 mL of distilled water, and heated in a water bath at 99 °C for 30 min before cooling to 25 °C. A total of 25 mL of the extract was added to distilled water, followed by 0.1 mL of the supernatant and 0.5 mL of 5% (w/v) salicylic acid in concentrated $H_2SO_4$ combined with (SA-$H_2SO_4$). After 20 min, the sample was cooled at room temperature, and 9.5 mL of 8% NaOH was gradually added to the combined supernatant. The absorption of the nitrogen content in the leaves at 410 nm was measured using spectrophotometer UV-5100B (Unico, Shanghai, China). The following formula was used to calculate the nitrate content (*Muneer et al., 2014*).

$$\text{Nitrate content } (\text{mg kg}^{-1} \text{ FW}) = \frac{\text{C} \times \text{V}_{\text{T}}}{\text{W} \times \text{V}_{\text{S}}} \tag{4}$$

where: C = the nitrate value from the standard curve ($\mu$g m L$^{-1}$); Vt = Total samples volume extracted; $V_s$ = Taken sample solution (4 mL); W = Fresh leaf weight (g).

### Soluble protein content

The soluble protein content of lettuce was measured using the Coomassie brilliant blue G-250 dye method adopted in *Muneer et al. (2014)*. Liquid nitrogen and 5 mL of distilled water were also used to grind 0.5 g of fresh lettuce into pulp. After centrifuging the extract solution at 11,200 $g$ for 10 min at 4 °C, 0.05 mL supernatant was combined with 0.95 mL distilled water and 5 mL Coomassie brilliant blue G-250 solution (0.1 g L$^{-1}$; Sigma-Aldrich, St. Louis, MO, USA). After 2 min, the absorption was measured at 600 nm using a UV-5100B spectrophotometer (Unico, Shanghai, China).

### Soluble sugar content

Fresh leaves were cut into small pieces and weighed (0.2 g) to measure some biochemical compounds of interest. The soluble sugar content was calculated using the anthrone colorimetric method (*Fu et al., 2012*). The absorbance of the extracted solution was

measured using a UV-5100B spectrophotometer at 595 nm (OD595) (Unico, Shanghai, China).

### Ascorbic acid content and total nitrogen utilization efficiency

RQFlex plus reflectometer (Merck, Darmstadt, Germany) was used to determine the ascorbic acid content of plants in each treatment, according to the procedure leveraged by *Tabata et al. (2001)*. The following formula was used to compute nitrogen-utilization efficiency (NUtE) (*Siddiqi et al., 1990*), with the results reported as g 2DW mg$^{-1}$ N:

$$\text{NUtE} = \frac{\text{Total leaves DW}}{\text{Total nitrogen content TNC}}. \qquad (5)$$

### Photosynthetic productivity and efficiency

The measurements were taken as previously described in *Guiamba et al. (2022)* for the following parameters: net photosynthetic rate (A), stomatal conductance (gs), leaf transpiration rate (Tr), and intracellular $CO_2$ concentration (Ci). These parameters were used as indicators of the plant's gas exchange performance. The outer and inner leaves (fully developed leaves) were randomly sampled from each plant's top. Each duplicate had four leaves from four separate plants (16 replicates). Leaf temperatures were maintained at 25 °C, $CO_2$ concentrations were maintained at 400 μmol mol$^{-1}$, and relative humidity (RH) was maintained at 70%.

### Electricity consumption measurements

An electricity meter (LCBG-ZJ341-07; LiChuang Science and Technology Co., Laiwu, China) was utilized to calculate the illumination's electricity consumption (the energy costs for cooling, ventilation and the recirculation of the nutrient solution were not considered during our experiments). This tool was also used to determine how much light each treatment used (LUE (grams per kilowatt hour)) (LUE = leaf FW (grams per plant) 18 plants/m$^2$/electric-energy consumption of illumination (kilowatts per hour)).

## Statistical analysis

Data were analyzed using "Statistix 8.1" (*Gomez & Gomez, 1984*; *Sánchez-Rodríguez et al., 2010*). We performed t-test to find significant differences among treatments.
The significant differences between treatments were compared by the least significant difference LSD ($p \leq 0.05$) (*Halberstadt et al., 2008*).

## RESULTS

### Plant growth

After cultivating the plants under the N-S and the C-S in the plant factory, we observed distinct variations among the various parameters assessed. For instance, the C-S treatment had a significant impact on the morphology of lettuce plants, including the length of the shoots (Fig. 6A), the diameter of the stems (Fig. 6B), and the number of leaves (Fig. 6D). When the supplement adjustable sideward lighting LEDs were switched on and used in conjunction with the movable downward lighting LED system, root fresh weight between
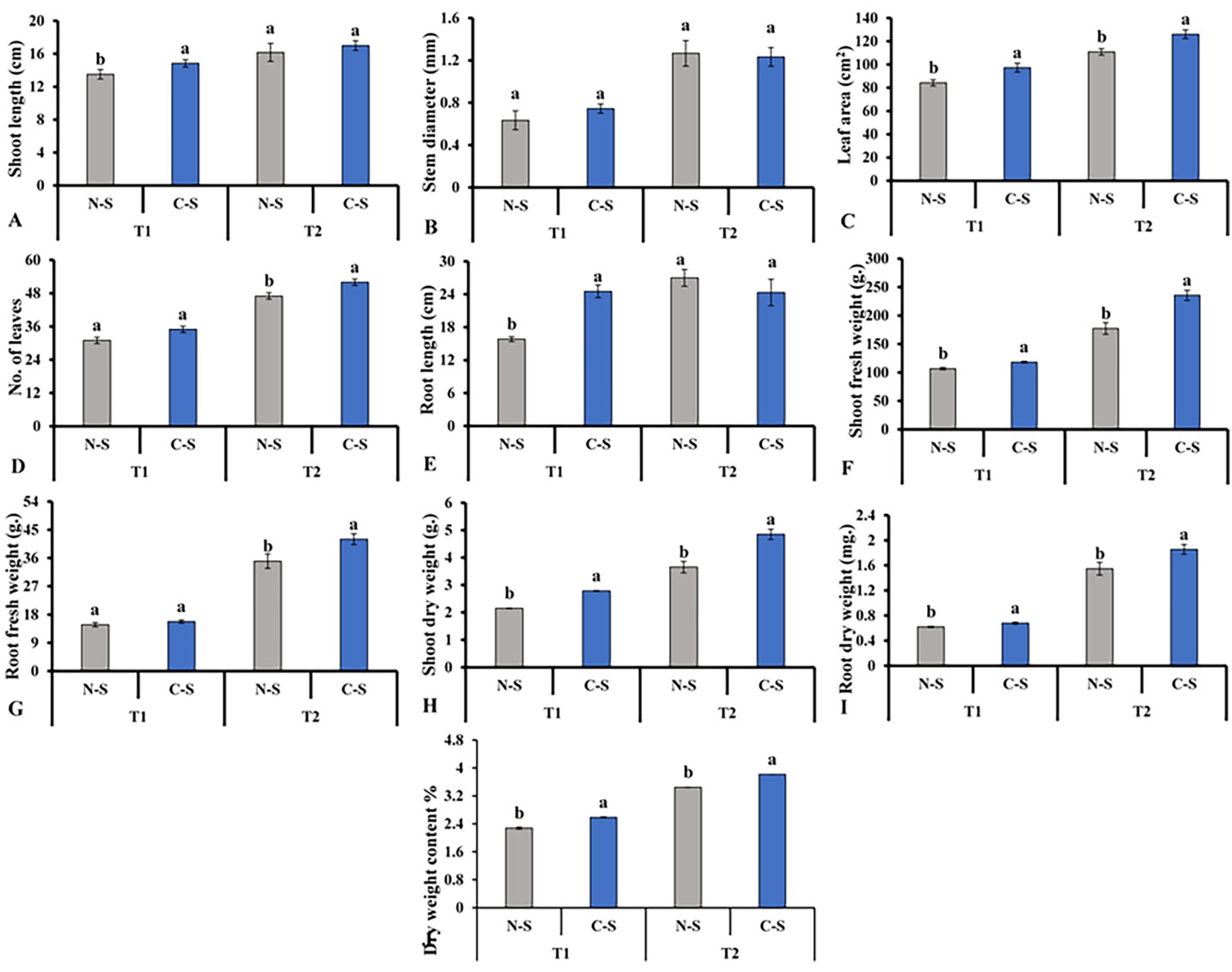

**Figure 6  Effects of different illumination systems.** Effects of different illumination systems on shoot length (A), stem diameter (B), leaf area (C), number of leaves (D), root length (E), shoot fresh weight (F), root fresh weight (G), shoot dry weight (H), root dry weight (I), and dry weight content (J). Values are means of four technical replicates ($n = 5$) ± standard error; $^{*}$ = is significant at $p \leq 0.05$ level; $^{**}$ = is significant at $p \leq 0.01$ level for each time between treatments according to the t-test.

N-S and C-S revealed no significant difference (Fig. 6G). However, the leaf area, shoot fresh weight, shoot dry weight, root dry weight content under C-S were significantly higher than N-S at T2, as shown in Figs. 6C, 6E, 6F, 6H–6J, respectively.

The plants treated under the C-S and the N-S had very different morphologies (Fig. 7). For example, plants subjected to the C-S appeared dense, with narrow, twisted leaves, and no excessive elongation was observed. On the other hand, plants under the N-S were short with hypertrophic and thick leaves and a yellowing color, despite it been subjected to light condition for 40 days, and exhibited a sparsest plant architecture with a noticeable elongation of the stem. This implies that the optimal wavelength for plants should be adjusted following plant growth and that the C-S exhibited superior performance.

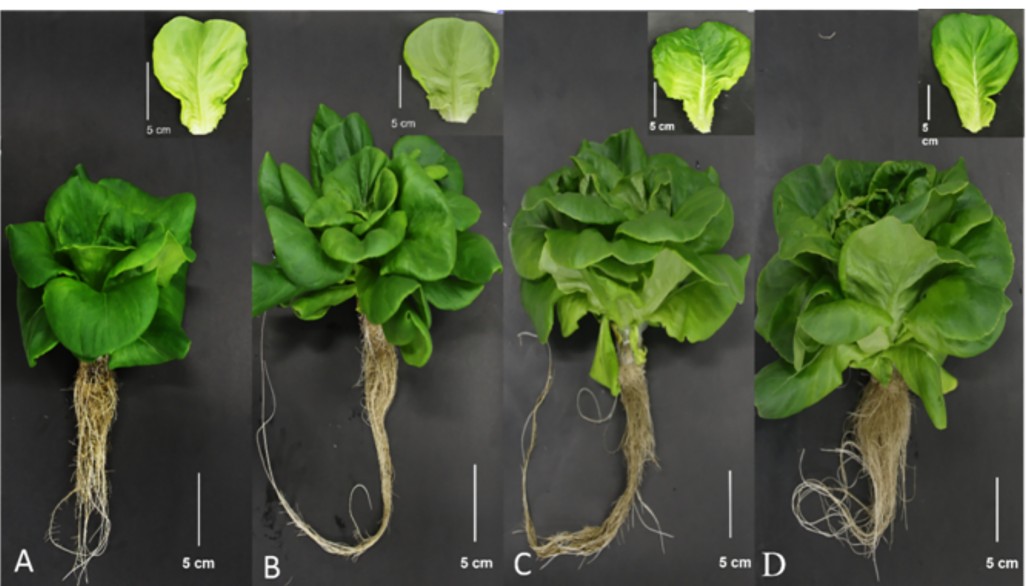

**Figure 7 Morphology.** Morphology (N-S at T1 (A), C-S at T1 (B), N-S at T2 (C), and C-S at T2 (D)) of lettuce (at harvest) planted in varied light treatments.     

## Photosynthetic pigments content and biochemical contents

The supplementary sideward light source significantly impacted the levels of chlorophyll *a* and *b*, soluble sugar, and soluble proteins. The lettuce grown under C-S at T2 had the higher chlorophyll *a* and *b* concentrations (Figs. 8A and 8B). It was observed that chlorophyll *a* concentration under N-S at T1 was the lowest compared with C-S at T2. At T2, lettuce's carotenoid concentration dropped under C-S largely due to the additional sidelight compared to plants cultivated under N-S. This indicates that time directly affects the transformation of carotenoids into pigment in plants, as shown in Fig. 8C.
The supplemental lights helped to increase the amount of soluble sugar in the lettuce leaves, with C-S at T2 yielding the higher level compared with N-S (Fig. 8F). The N-S-treated lettuce had the lowest soluble protein levels at T1, but these levels somewhat increased at T2 relative to those under C-S. The highest soluble protein levels were found at T2 under C-S compared with N-S (Fig. 8E). However, nitrate content exhibited no discernible variations between the two treatments (Fig. 8D).

## Ascorbic acid content and total nitrogen utilization efficiency

The ascorbic acid concentrations in plants under the C-S treatment varied significantly compared with those under the N-S treatment. The outer leaf revealed a generally increase in ascorbic acid level than the inner and total leaf (Fig. 9A). Total nitrogen utilization efficiency in plants under the C-S was higher than that under N-S at both T1 and T2 (Fig. 9B).

## Photosynthetic productivity

It was noticed that the C-S treatment influenced the photosynthetic rate, stomatal conductance, evaporation rate, and $CO_2$ concentration in the newest fully developed leaves
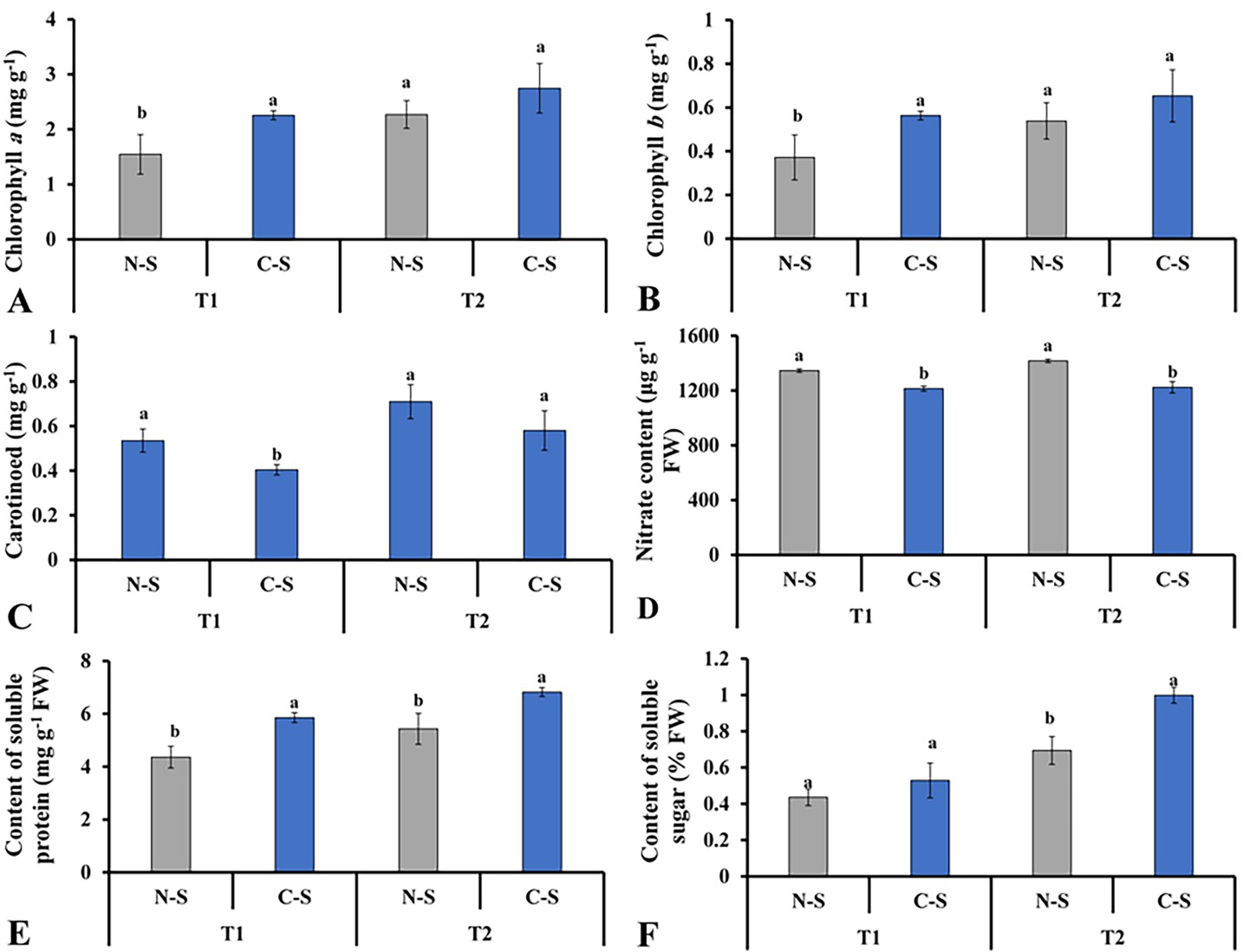

**Figure 8 Effect of different illumination systems on the photosynthetic pigments content and biochemical contents.** Chlorophyll *a* (A), chlorophyll *b* (B), carotenoid (C), nitrate content (D), content of soluble protein (E), content of soluble sugar (F). Values are means of four technical replicates ($n = 5$) ± standard error; * = is significant at $p \leq 0.05$ level; ** = is significant at $p \leq 0.01$ level for each time between treatments according to the t-test.

with the additional adjustable sideward lighting at a PPFD of 50 μmol m$^{-2}$ s$^{-1}$ compared with the N-S treatment. The lowest rate of evaporation was observed in the control group, while stomatal conductance showed similar patterns. Under the C-S, which was about twice that of the control, the stomatal conductance peaked significantly. The leaves showed positive net photosynthetic rates (Table 2). These findings showed that leaves' ability to increase photosynthesis could be regulated by additional, adjustable side lighting.

## Light and energy use efficiency

The segmented energy consumption of the C-S and N-C is shown in Table 3. C-S exhibited an upward trend on day 20, while N-S performed in terms of energy consumption until the harvesting period. The power consumption per unit area, the output per unit area, and the

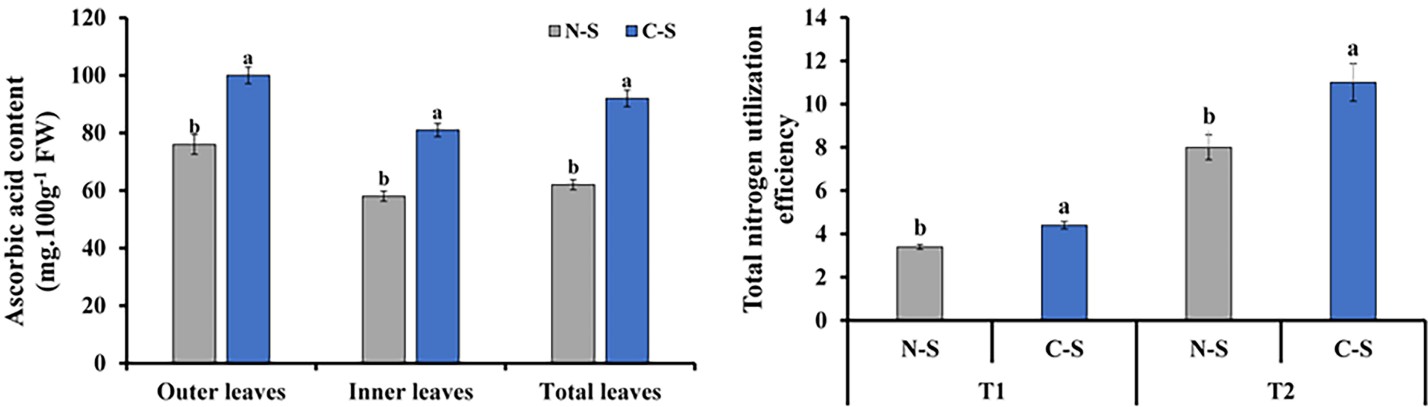

**Figure 9 Ascorbic acid content and total nitrogen utilization efficiency.** (A) Ascorbic acid content at 40 days after transplanting in the outer leaves, inner leaves, and total leaves of lettuce plants grown under N-S and C-S. (B) Total nitrogen utilization efficiency cultivated under N-S and C-S at T1 and T2 after transplanting. Values are means of four technical replicates ($n = 5$) ± standard error; * = is significant at $p \leq 0.05$ level; ** = is significant at $p \leq 0.01$ level for each time between treatments according to the t-test.

**Table 2 Effects of different illumination systems on net photosynthetic rate (A), stomatal conductance ($g_s$), leaf transpiration rate ($Tr$), and intracellular $CO_2$ concentration ($C_i$).**

| Photosynthetic parameters | The experimental group (C-S) | Control group experiment (N-S) |
|---|---|---|
| Net photosynthetic rate ($\mu$mol $CO_2$ m$^{-2}$ s$^{-1}$) | 6.50 ± 0.34a | 5.72 ± 0.38b |
| Stomatal conductance (mol $H_2O$ m$^{-2}$ s$^{-1}$) | 0.09 ± 0.10a | 0.05 ± 021b |
| Evaporation rate ($\mu$mol $H_2O$ m$^{-2}$ s$^{-1}$) | 0.53 ± 0.07a | 0.37 ± 0.11b |
| $CO_2$ concentration ($\mu$mol $CO_2$ mol$^{-1}$) | 305.10 ± 24.54a | 282.43 ± 12.63b |

**Table 3 Segmented energy consumption of different lighting systems.**

| Segmented energy consumption (kWh) | (C-S) | (N-S) |
|---|---|---|
| 1–5 days | 6.5 | 9.8 |
| 5–10 days | 13 | 19.6 |
| 10–15 days | 19.5 | 29.4 |
| 15–20 days | 26 | 39.2 |
| 20–25 days | 32.15 | 49 |
| 25–30 days | 39 | 58.8 |
| 30–35 days | 44.8 | 68.6 |
| 35–40 days | 52.25 | 80.41 |

power utilization efficiency (LUEs) of the C-S and the N-S were measured and calculated separately (Table 4). The comparison showed that the power consumption of the C-S was (35.02%) lower than that of the N-S. We pointed out that the power utilization rate of LUEs was a practical value for comparing the power utilization rate between the systems. Therefore, it was observed that at relatively low output and the power utilization rate of the C-S were (135%) higher than that of the N-S. The analysis also indicated that C-S can allow

**Table 4 Effects of different illumination systems on electricity consumption, plant yields, and efficiency LEDs.**

| Growth parameter(s) | (C-S) | (N-S) |
|---|---|---|
| E-consumption (Kw h m$^{-2}$) | 52.25 | 80.41 |
| Plant yields (g m$^{-2}$) | 5,150.04 | 3,937.73 |
| Efficiency LEDs (g kw h$^{-1}$) | 93.92 | 40.47 |

the lettuce to receive the ideal light under the conditions of large-scale plantations in plant factories, obtain relatively more biomass, and consume the least energy value.

## DISCUSSION

### Response of the morphology and photosynthetic pigment under different irradiances

Plant's morphological traits can be influenced by a number of factors, and they have developed mechanisms to adapt to various environments (*Kumar et al., 2018*; *Li et al., 2021*). Research has demonstrated that lighting quality, intensity, source, and photoperiod affect plant morphology. However, only a few studies have investigated the orientation of the light source and how it influences the development and growth of plants. *Joshi et al. (2017)* demonstrated that lettuce grows more efficiently under a combination of downward and upward lighting settings. Moreover, high light levels boost photosynthesis and subsequently increase plant biomass, whereas low light intensities typically result in photoinhibition and impact plant photomorphogenesis (*Chen et al., 2016*; *Kelly et al., 2020*). *Woltering & Witkowska (2016)* showed that higher PPFD increased the dry matter content of lettuce, indicating higher carbohydrate levels and improved post-harvest quality. Here, we observed that C-S significantly outperformed N-S in terms of above-ground and root biomass (fresh weight and dry weight), dry matter content, and leaf weight of lettuce while significantly delaying the senescence of outer leaves in plants, thus leading to a higher photosynthetic activity. These results aligned with recent studies, where in plant shoots' fresh weight/dry weight was influenced by both plant morphology and photosynthetic capacity (*Hernández & Kubota, 2015*; *Wang et al., 2016*). However, tip burn was observed among lettuce plants cultivated indoors under the increased light intensity. Correspondingly, *Sago (2016)* noted that butterhead lettuce frequently experienced tip burn when exposed to high light intensities (150 and 300 μmol·m$^{-2}$·s$^{-1}$), which may have been induced by a calcium deficit in the interior leaves (*Sago, 2016*). The lettuce cultivar used in the current study were grown in C-S and N-S treatments with higher PPFD of 200 + 50 and 280 μmol·m$^{-2}$·s$^{-1}$, respectively. Neither cultivar displayed the tip burn phenomenon, proving that these PPFD were not excessive for lettuce growth.

Plants rely heavily on chlorophylls (Chl) for photosynthetic capacity and growth (*Li et al., 2018*). Plant chlorophyll content is also significant for the visual appearance of the vegetable. A consumer will accept or reject a product based on its color and appearance, and these qualities are even more important in a product, such as microgreens, which are

highly valued for the colors they include (*Barrett, Beaulieu & Shewfelt, 2010*). Chlorophyll levels can be influenced by both the intensity of the illumination and the plant's genotype, as has been described in previous studies (*Teng, Liao & Wang, 2021*). In lettuce treated with 250 µmol m$^{-2}$s$^{-1}$, it was evident that higher light intensities were easily accessible to higher chlorophyll *a/b* levels. In contrast, the concentration of chlorophyll *a* and *b* decreased by 300 µmol·m$^{-2}$·s$^{-1}$. Likewise, the use of 200 µmol m$^{-2}$s$^{-1}$+ 50 µmol m$^{-2}$s$^{-1}$ under C-S at T2 was beneficial for the accumulation of chlorophyll *a* and *b* (Figs. 8A and 8B), and this enhancement was discovered to be directly associated with the development of the leaf structures.

The senescence of the leaves is a fundamental attribute that decreases the post-harvest performance and the nutritional content of horticultural crops, the yield of agricultural crops, and the build up of biomass. Senescence also restricts the production of agricultural crops (*Guo & Gan, 2014*). Furthermore, it can be prolonged by a lack of light (*Guo & Gan, 2014*; *Bresson et al., 2017*). On the other hand, light regulation remains a major issue in large-scale plantations (high plant density) due to occlusion caused by adjacent plant leaves, making it difficult for the middle and lower leaves to receive adequate light compared to the upper leaves (*Chapepa, Mudada & Mapuranga, 2020*). Correspondingly, we observed that lettuce grown in the N-S treatment exhibited yellow color at T2. This result could be ascribed to the unbalanced radiation on the cultivar plate, which accelerates the senescence of leaves.

In contrast, lettuce grown in the C-S treatment tent at T2 displayed an appropriate green hue. The senescence of leaves due to low light levels was accompanied by the transfer of nutrients to younger tissues, the loss of chlorophyll, the breakdown of photosynthetic proteins, and a decrease in photosynthetic activity (*Thimann & Satler, 1979*; *Wingler et al., 2006*). These findings demonstrated that the sideward and upward combination might delay the middle leaf's senescence in plant factories. This finding is consistent with the results documented in previous studies, in which it was pointed out that supplemental lighting system delayed leaf's senescence of lettuce (*Zhang et al., 2015*; *Joshi et al., 2017*). This result explains the increment in marketable leaves, as shown in Fig. 10A.

## Response of soluble protein, nitrate, and soluble sugar under different irradiances

An accumulation of soluble carbohydrates could result from more intense lighting (*Bian, Yang & Liu, 2015*). The soluble protein plays a vital role in the osmotic regulation and metabolism of several metabolic enzymes. Plant resistance and metabolism are embodied in soluble protein content. As a result of the synergistic effect of the C-S lighting's improved stomatal characteristics, photosynthetic pigment concentrations, and use efficiency of light delivered in diverse directions, we observed that the soluble protein levels were increased under C-S lighting at T2 (Fig. 8F).

Nitrates are one of the most important molecules in determining the nutritional value of food. Nitrate ($NO_3^-$) can accumulate to varying degrees in vegetables and may cause human health issues (*Kim, Chiami & Ishii, 2006*). Romaine lettuce is a strong provider of nutrients and has low $NO_3^-$. Many factors, including light levels, contribute to the final

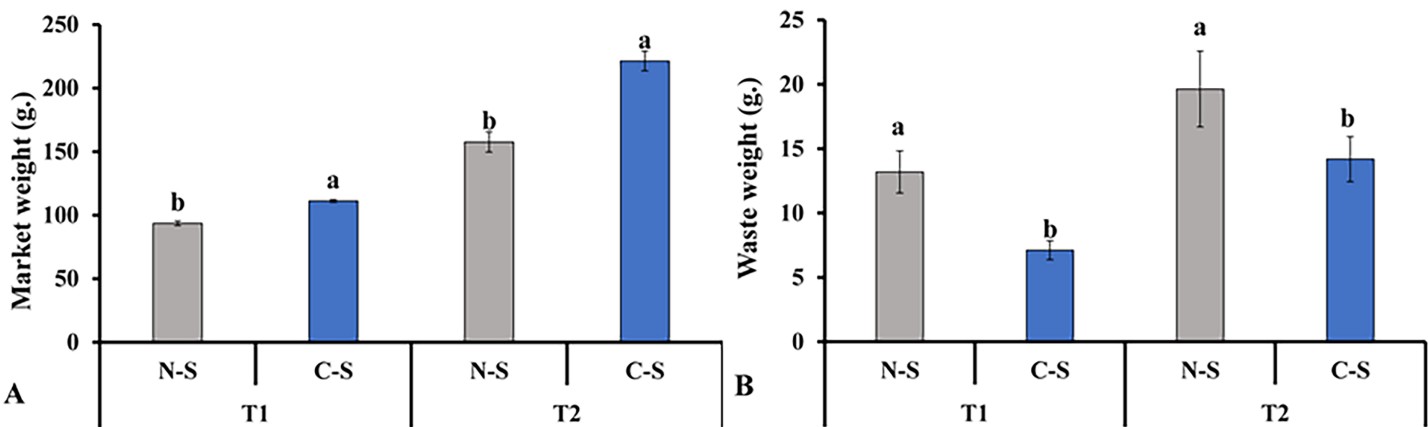

**Figure 10 Total marketable leaves fresh weights and waste leaves.** Total marketable leaves fresh weights (A) and waste leaves (B) from plants cultivated under N-S and C-S at T1 and T2 after transplanting. Values are means of four technical replicates ($n = 5$) ± standard error; * = is significant at $p \leq 0.05$ level; ** = is significant at $p \leq 0.01$ level for each time between treatments according to the t-test.

nitrate content in plants (*Lillo, 2007*). The light intensity and quality affect nitrate uptake, translocation, and decrease in plant organs. Romaine lettuce grown under light-emitting diodes with changing the light conditions during growth had lower levels of nitrate (*Nicole et al., 2018*). Our results showed that nitrate concentrations were higher under N-S, with a substantial impact from the substrate component. Our finding partly agree with *Jannat et al. (2022)* study, where in it was established that nitrate levels in coconut fiber dropped dramatically due to exposure to supplemental light (Fig. 8D).

It has been established that soluble sugars are highly sensitive to environmental stresses. They act on the supply of carbohydrates from source organs to sink organs. Besides, regulating lettuce biomass and morphology and light intensity also impacts nutrition quality and anti-oxidative enzyme activities (*Fu et al., 2012*; *Fan et al., 2013*). Light intensity increases lettuce's soluble sugar and ascorbic acid content (*Zhou et al., 2012*). Furthermore, different lighting conditions that exist throughout the growing cycle have the potential to influence the total amount of carbohydrates present in the plant. These variables can stimulate photosynthetic activity (*Cocetta et al., 2017*). Vegetables' sucrose concentration and nutritional profile can be improved with the help of a special LED light, as revealed by *Lin et al. (2013)*. It was discovered that the amount of soluble sugar in strawberries rose in proportion to the direction of the irradiance when the top and side illumination were utilized (*Yang, Song & Jeong, 2021*, *2022*). In a related study, there was an increase in the levels of expression of genes involved in sugar metabolism and signaling in lettuce cultivated with supplemental lighting (*Viršilė et al., 2019*). Here, we observed that soluble sugar content performed better under the C-S at t2. This could be attributed to usage of the supplemental adjustable sideward lighting and the change in irradiance direction. These results were consistent with prior research, indicating that the principal lighting direction can enhance the concentration of primary metabolites (*Yang, Song & Jeong, 2021*, *2022*).

### Response of ascorbic acid under different irradiances

Vitamin C, also known as ascorbic acid, is found in high concentrations in plant foods. It is present in all cell compartments, including the cell wall, and reaches a concentration of more than 20 mM in the chloroplasts where it is found (*Smirnoff & Wheeler, 2000*; *Gallie, 2013*). Multiple reports indicate light enhances vitamin C concentration in plants (*Li et al., 2010*; *Massot et al., 2011*). The varied light irradiance impacted ascorbic acid in romaine lettuce in the C-S and N-S (Fig. 9). The ascorbic acid content of plants growing under C-S was higher than under N-S. Similar results have been reported for ascorbic acid, alpha-carotene, and phenolic compound levels when treated with UV light (*Xie et al., 2015*) or red light (*Bliznikas et al., 2012*) during the latter stages of culture.

### Response of stomata under different irradiances

The opening and closing of guard cells control the quantity of stomata, which carry air and water vapor during carbon assimilation, respiration, and transpiration. Light, temperature, and $CO_2$ are exogenous factors influencing the stomata's opening and closing. In our study, the supplemental sideward lighting enhanced the stomatal opening compared with the N-S lighting. Moreover, romaine lettuce plants grown under C-S lighting appear to have higher photosynthetic efficiency due to favorable stomatal conditions (*Kardel et al., 2009*; *Moore et al., 2021*). We, therefore, believed that this phenomenon led to the highest biomass production rates, plant growth, and development. This result is in accordance with previous findings, where in it was established that side lighting could promote stomatal formation (*Yang, Song & Jeong, 2021*).

### Electric-energy consumption of the lighting systems

Optimizing the plant factories' performance across their energy spectrum is crucial for ensuring the design's long-term viability. Lighting fixture position influences cultivation productivity and system lighting efficiency LED panels with different illumination schedules and mounted above butterhead lettuce (*Lactuca sativa* capitata) seedlings were examined. It was revealed that the highest light efficiencies and lowest electricity consumption were observed in the treatments with irradiation from a shorter distance above the seedlings (*Li et al., 2014*). In this study, the energy usage fell by 35.02%, and the power utilization rate increased by 135% when the LED panels were mounted adjacent and above the plants, suggesting thaty lighting system built in this study has energy-saving properties and a high energy utilization rate in the seedling stage of plant growth. This result is consistent with *Cocetta et al. (2017)* research, in which they found that reducing fixture spacing and focusing lights at the canopy could minimize lighting pollution and energy consumption, and lightning-quick efficiency.

## CONCLUSIONS

The results of our study clearly show that the two systems had different effects on lettuce culture in the plant factory at T1 and T2 under different lighting irradiance. We also proved that the new lighting type systems successfully reduced the quantity of light energy consumption in the plant factory and boosted lettuce yields and quality. Moreover, this

system moderated the decrease of chlorophylls and prevented premature aging, and falling of lower leaves. C-S lighting also proved to be a system that can increase plant output through a correct light source configuration and make the plant factory system applicable to many plants. We, therefore, concluded that using C-S lighting systems in plant factories that use artificial lighting will result in significant benefits. However, more research is required about the onset of supplemental sideward and adjustment changes in response to different lighting directions.

### Funding
This work was financially supported by the Nature Science Foundation of Fujian Province of China (2017J01423). The funders had no role in study design, data collection and analysis, decision to publish, or preparation of the manuscript.

### Grant Disclosures
The following grant information was disclosed by the authors:
Nature Science Foundation of Fujian Province of China: 2017J01423.

### Competing Interests
The authors declare that they have no competing interests.

### Author Contributions
- Mulowayi Mutombo Arcel conceived and designed the experiments, performed the experiments, analyzed the data, prepared figures and/or tables, authored or reviewed drafts of the article, and approved the final draft.
- Ahmed Fathy Yousef performed the experiments, prepared figures and/or tables, and approved the final draft.
- Zhen Hui Shen analyzed the data, authored or reviewed drafts of the article, and approved the final draft.
- Witness Joseph Nyimbo performed the experiments, analyzed the data, prepared figures and/or tables, and approved the final draft.
- Shu He Zheng conceived and designed the experiments, performed the experiments, analyzed the data, authored or reviewed drafts of the article, and approved the final draft.

### Data Availability
The raw data is available as a Supplemental File.

### Supplemental Information
Supplemental information for this article can be found online at http://dx.doi.org/10.7717/peerj.15401#supplemental-information.

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
