# Peer review of "Optimizing lettuce yields and quality by incorporating movable downward lighting with a supplemental adjustable sideward lighting system in a plant factory"

_PeerJ, doi:10.7717/peerj.15401_

## Round 0.1 · original submission · Major Revisions

The experiment design and discussion parts need to be tailored. Please revise this manuscript according to each comment made by the reviewers. Besides, please carefully check the whole manuscript to avoid small but unnecessary typos that should not appear in a submitted manuscript. For example, in line 107, it should be ‘PPFD’ instead of ‘PPDF’; in line 342, it should be ‘N-S’ instead of ‘N-C’.

Reviewer 1 ·

Basic reporting

Introduction & background were according to context. Literature was well referenced & relevant. Figures are relevant, high quality, well labelled & described. However following suggestions were made to improve quality;
• Title of article should be more precise and technical
• Language quality needs improvement by professional.
• Problem statement is not much clear.
• Introduction is unnecessarily lengthy. Precision is required.
• The ultimate aim of the research needs clarity.

Experimental design

EXPERIMENTAL DESIGN Original primary research within Scope of the journal. Research question well defined, relevant & meaningful. It is stated how the research fills an identified knowledge gap. However the experimentation part is also very lengthy and can be shorten.

Validity of the findings

All underlying data have been provided; they are robust, statistically sound, & controlled. Conclusions are well stated, linked to original research question & limited to supporting results.
• In results sections numeric data must be written in brackets where you are suggesting high or low comparisons
• Commercialization aspect of this can be added to the conclusion part.
• Discussion part is with plenty of unnecessary details that must be omitted to make it brief, concise.

·

Basic reporting

Appropriate

Experimental design

Appropriate experimental design with respect to the objectives of the study

Validity of the findings

Findings of the study are valid and applicable to a wider range of the readers particularly those who intend to use technologically advanced methods for the production of leafy vegetables.

Additional comments

At some places there is space between the units and the figures (for example line 169) and some place there is no space between these (for example line 170). Please see the whole manuscript and change/stick to one style throughout manuscript.
 Line 188 “mS/cm-1’ is duplication for per cm. Either write mS/cm or mS cm-1.
 Please replace ‘ml’ with ‘mL’ in whole manuscript.
 At some places there is space between the figure and its number (for example line 317 ‘Figure 8 A and B’) and some place there is no space between these (for example line 321 Figure ‘8C’). Please see the whole manuscript and change/stick to one style throughout manuscript.
 Conclusion is a sort of summary. It should be a take home message out of study and should be concisely written in 3-4 lines. It is recommended to re-write the conclusion.
 I have not checked the references, but in a bird’s eye view many contradictions in the style/format of reference can be seen, which need to be fixed.
 It is very difficult to understand Figure 1. It will be better to give the sketch of figure 1 from different angles so that readers can have better understanding of it.
 ‘t-test’ is written with small ‘t’ not with capital ‘T’. Please change in all the figure legends and whole manuscript.
 Please add the probability values in all the figures.
 Legends of figures 10 and 11 are incomplete and * and ** are missing, please re-check and correct.

·

Basic reporting

With the manuscript 'Way to boost lettuce yields and quality by incorporating movable downward lighting with a supplemental adjustable sideward lighting system in a plant factory', authors Arcel et al. describe an interesting concept on adjustable lighting systems. However, the manuscript has significant flaws.

- The English language is lacking and needs to be improved dramatically to be able to properly understand the authors findings.
- There are many errors in the literature references both in formatting and referencing the wrong sources.
- Some figures lack proper labels or even the wrong figure is shown. Image quality is also lacking.

Please refer to the annotated pdf for more details.

Experimental design

Please refer to the annotated pdf for more details.

Validity of the findings

When describing the data in the results, often statements are made that do not coincide with the data provided in the figures, which is not acceptable.

Please refer to the annotated pdf for more details.

Reviewer 4 ·

Basic reporting

Overall, the quality of this manuscript is very good. well written and well described with critical scientific manners.
references, background, structure, figure all are good.

Experimental design

Its very good. authors clearly described the methodology.
It is a original work with sufficient investigation

Validity of the findings

well written. data are meaningful and the conclusion is OK.

Additional comments

I highly appreciate it being published.
As a new journal, this type of article must add worth to this journal.

---

## Round 0.2 · accepted · Accept

Congratulations on the acceptance of your manuscript! I appreciate the authors' hard work and dedication during the review process.

The Section Editor noted:

> There are a few typos that need to be corrected. Please spell out acronyms such as N-S and C-S in the figure captions.

Reviewer 1 ·

Basic reporting

All the corrections and suggestions are incorporated in the manuscript. iam satisfied with the improvement in the manuscript. I have no hesitation to accept it.

Experimental design

All the corrections and suggestions are incorporated in the manuscript.iam satisfied with the improvement in the manuscript. I have no hesitation to accept it.

Validity of the findings

All the corrections and suggestions are incorporated in the manuscript.iam satisfied with the improvement in the manuscript. I have no hesitation to accept it.

Additional comments

All the corrections and suggestions are incorporated in the manuscript.iam satisfied with the improvement in the manuscript. I have no hesitation to accept it.